# Protective Role of Bergamot Polyphenolic Fraction (BPF) against Deltamethrin Toxicity in Honeybees (*Apis mellifera*)

**DOI:** 10.3390/ani13243764

**Published:** 2023-12-06

**Authors:** Roberto Bava, Fabio Castagna, Stefano Ruga, Rosamaria Caminiti, Saverio Nucera, Rosa Maria Bulotta, Clara Naccari, Domenico Britti, Vincenzo Mollace, Ernesto Palma

**Affiliations:** 1Department of Health Sciences, University of Catanzaro Magna Graecia, 88100 Catanzaro, Italy; roberto.bava@unicz.it (R.B.); fabiocastagna@unicz.it (F.C.); rugast1@gmail.com (S.R.); rosamariacaminiti4@gmail.com (R.C.); saverio.nucera@hotmail.it (S.N.); rosamaria.bulotta@gmail.com (R.M.B.); c.naccari@unicz.it (C.N.); britti@unicz.it (D.B.); mollace@unicz.it (V.M.); 2Mediterranean Ethnobotanical Conservatory, Sersale (CZ), 88054 Catanzaro, Italy; 3Department of Health Sciences, Institute of Research for Food Safety & Health (IRC-FSH), University of Catanzaro Magna Graecia, 88100 Catanzaro, Italy; 4Nutramed S.c.a.r.l., Complesso Ninì Barbieri, Roccelletta di Borgia, 88021 Catanzaro, Italy

**Keywords:** honeybee (*Apis mellifera*), natural products (NPs), bergamot polyphenolic fraction (BPF), deltamethrin, pyrethroid

## Abstract

**Simple Summary:**

The inappropriate use of pesticides is one of the underlying causes of the extensive honeybee die-off that is taking place worldwide. Pesticides can cause acute, sub-acute, and chronic toxicity, in the latter case leading to physiological and behavioral changes in honeybees. Acute pesticide toxicity is exacerbated by nutritional deficiencies and poor immune system fitness. Studies have shown that polyphenols extracted from many botanical species have been shown to possess protective properties against acute and chronic pesticide intoxication. In this study, we wanted to test the protective role of the bergamot polyphenolic fraction (BPF) against deltamethrin intoxication in honeybees. The semi-filed toxicity tests showed that, when administered in combination with deltamethrin, BPF was able to protect against pesticide toxicity. In particular, survival indices were improved, and the honeybees showed a reduction in abnormal behavior compared to the positive control group treated with deltamethrin alone.

**Abstract:**

Pesticide-induced poisoning phenomena are a serious problem for beekeeping and can cause large losses of honeybee populations due to acute and sub-acute poisoning. The reduced responsiveness of honeybees to the damage caused by pesticides used in agriculture can be traced back to a general qualitative and quantitative impoverishment of the nectar resources of terrestrial ecosystems. Malnutrition is associated with a decline in the functionality of the immune system and the systems that are delegated to the detoxification of the organism. This research aimed to verify whether bergamot polyphenolic extract (BPF) could have protective effects against poisoning by the pyrethroid pesticide deltamethrin. The studies were conducted with caged honeybees under controlled conditions. Sub-lethal doses of pesticides and related treatments for BPF were administered. At a dose of 21.6 mg/L, deltamethrin caused mortality in all treated subjects (20 caged honeybees) after one day of administration. The groups where BPF (1 mg/kg) was added to the toxic solution recorded the survival of honeybees by up to three days. Comparing the honeybees of the groups in which the BPF-deltamethrin association was added to the normal diet (sugar solution) with those in which deltamethrin alone was added to the normal diet, the BPF group had a statistically significant reduction in the honeybee mortality rate (*p* ≤ 0.05) and a greater consumption of food. Therefore, it can be argued that the inclusion of BPF and its constituent antioxidants in the honeybee diet reduces toxicity and oxidative stress caused by oral intake of deltamethrin. Furthermore, it can be argued that BPF administration could compensate for metabolic energy deficits often induced by the effects of malnutrition caused by environmental degradation and standard beekeeping practices.

## 1. Introduction

The honeybee die-off that has occurred in recent years is due to numerous factors [1,2]. Among these, pesticide poisoning is particularly important [3,4]. The term pesticides is generally used to indicate all products that are employed to combat harmful organisms, especially in agriculture [5]. Unfortunately, it is not only the organisms targeted by pesticide treatments that suffer but very often also the useful ones [6,7]. Honeybees can be exposed to pesticides through a variety of routes, with contact or ingestion being the main two [8]. In addition to the usual way, contact between honeybees and pesticides can also occur in ways that are not easily predictable. For example, the treatment of the seeds of various crops with some insecticides, in particular neonicotinoids, can pose a danger to honeybees comparable to that of spray formulations [9]. During sowing, the abrasion of the film covering the seed in the hopper of the seeder produces dust that is dispersed in the surrounding environment, which can form a deadly cloud for the bees in flight that cross it or cause, after being deposited, intoxications at a later time [9,10]. Contamination can also occur at a considerable distance from the treated field due to the drift effect caused by the wind or through indirect contamination [11,12]. According to Krupke et al. (2012) [13], the main sources of hive contamination for adults and larvae are nectar, recently stored pollen, and bee bread [13]. When forager honeybees collect contaminated pollen, nectar, honeydew, or water, they may die immediately, or if the dosages are not fatal, the contaminants may be carried into the hive. During the most crucial times, when honeybees need to utilize their food reserves, they are likely to consume these contaminants [13]. Therefore, moderate levels of pesticides can still pose a serious threat to the honeybee colony, leading to a gradual decline in the population. Paradoxically, direct treatments of honeybees with very toxic substances can have less impact, as by killing or preventing the foragers from returning to the hive, they do not act on the brood and the queen’s fecundity. In these cases, families can recover quickly by replacing the foragers, even if there is an irreversible loss of harvest. Most pesticides have different levels of toxicity based on a variety of variables, such as the exposure method, the age of the honeybees, the fitness of the colonies or honeybee subspecies, and the nutritional requirement [8,14,15]. The research studies revealed significant levels of pesticide toxicity and the related detrimental consequences for honeybees [16,17,18]. For example, research attention has been focused on neonicotinoid pesticides [19,20,21], and nowadays their use has been limited due to their negative effects on honeybees and growing resistance to them in treated pest populations [22].

Today, pesticides commonly used in agriculture belong to the class of pyrethroids [23,24]. Their insecticidal activity is due to the lipophilic keto-alcoholic esters of chrysanthemum and pyrethroic acid [25]. First- and second-generation pyrethroids are distinguished. Compared to second-generation (Type II) pyrethroids, first-generation (Type I) pyrethroids are less dangerous to mammals [26]. The voltage-dependent sodium channels are the major target of pyrethroid pesticides. Because of their low environmental persistence and toxicity, the pyrethroid family of insecticides is employed as alternatives to organochlorines, organophosphates, and neonicotinoids in pest-control programs [27,28]. Deltamethrin is a Type II pyrethroid characterized by a broad spectrum of activity that is widely used in professional agriculture, forestry, and hobby farming [29]. Its use is not free from the dangers of pollinating insects. Deltamethrin disrupts physiology in honeybees by causing memory disturbances, hypofertility, hypothermia, alterations in body and intestinal development, and altering normal dances and foraging activity [30,31,32,33]. It is also important to emphasize that pesticides exacerbate their effects when the body’s defenses are poor.

In the last few years, the modification of the landscape caused by human activity has led to a decrease in the quality of bee food [34]. As agriculture becomes more intensive, the landscape changes and the availability of bee food supplies progressively decreases, which reduces environmental sustainability. Consequently, the variety of flowering plants has decreased [35]. Low species diversity of flowering plants results in a reduction in the variety and availability of macro- and micro-elements in bee nutrition, which ultimately has a detrimental impact on honeybee populations [18]. Ineffective beekeeping techniques also contribute to nutritional deficiency; when winter supplies need to be replenished, bees are sometimes given little more than a solution of sugar and synthetic pollen replacements. The majority of the time, these food supplements are deficient in nutrients that are present in honeybees’ normal diets [36,37]. The number of honeybee colonies may be declining as a direct consequence of poor nutrition, which may also increase susceptibility to diseases and pesticides [38]. Many studies have demonstrated that the dietary consumption of phenolic compounds and flavonoids can increase the body’s ability to respond to pesticide injuries and pathogens [18,39]. Depending on food sources, the quantity and intake of phenolic compounds and flavonoids could change considerably [40]. The phenolic acids, flavonoids, and their derivatives are often abundant and diverse in honeybees’ natural diets [41], and it is these varying concentrations and ratios that affect the detoxifying effects [42]. Natural products (NPs) have been shown to have several properties [43,44,45,46,47]. Supplementing the diet with NPs and/or their compounds, such as polyphenols, could help honeybees respond better to the injuries that pesticides pose to their normal physiology. *Citrus Bergamia* Risso and Poiteau, commonly known as bergamot, is a shrub native to the Calabrian area of Southern Italy [48]. Its juice and albedo both exhibit a distinctive flavonoid and flavonoid glycoside profile. The composition and especially the high flavonoid concentration of bergamot set it apart from other citrus fruits [48]. Researchers have found that BPF, a polyphenol-rich fraction produced from bergamot juice and albedo, possesses antioxidant, anti-inflammatory, lipid-lowering, and hypoglycemic properties [49,50,51,52,53,54,55]. Bergamot requires particular cultivation conditions, such as alluvial, clayey, and calcareous soils. Calabria is the world’s largest producer. Over 90% of the world’s bergamot production comes from this region [56]. Among the flavonoids and flavonoid glycosides present in bergamot, there are neoerocitrin, neoesperidin, naringin, rutin, neodesmin, roifolin, and poncirin [53]. A total of 95% of flavonoids are represented by flavanones. It also features carbohydrates, pectins, and other compounds [53]. Our research aims to shed more light on the intricate and subtle impacts that pesticides could have on honeybee behavior and health. This study’s primary objective was to ascertain the impact of polyphenols on the mortality of honeybees poisoned by deltamethrin, one of the most popular pyrethroids. The polyphenol fraction of bergamot, also known as BPF, was chosen for the food supplementation. To assess the positive effect of polyphenol intake, the mortality rate of the honeybees fed BPF-deltamethrin in sugar solution was monitored over time and compared with that of honeybees taking only the pesticide in sugar solution. Furthermore, the abnormal honeybee behavior of the experimental groups as well as food consumption were assessed.

## 2. Materials and Methods

This investigation was carried out in the summer of 2023 at the Interregional Research Center for Food Safety and Health (IRC-FSH), Department of Health Science, University “Magna Graecia” of Catanzaro (Italy). The experimental honeybees came from one healthy honeybee colony. The honeybees used for the experiment belonged to the subspecies *A. mellifera ligustica*. The colony was conducted using conventional beekeeping methods. To ensure that the colony was healthy and free from parasitic or infectious infestations, established inspection procedures were used. As established by the standard methods for toxicological research, to obtain individuals of the same age, honeycombs of brood with hatching honeybees were placed in an incubator (35 °C and 65–80% relative humidity), and emerging honeybees were collected after 12 h [15].

After brushing the frames, all the honeybees were mixed and split into the experimental groups. Disposable cages have been used for toxicological studies because it is difficult to remove chemical residues [15]. Only one feeder (a plastic syringe with the tip cut off to enable solution flow) was given to each test cage [57].

Therefore, the young honeybees were moved into cages (20 per cage). To introduce the honeybees into the cages and administer the treatments, routine procedures without anesthesia were used [15,58]. After honeybees were introduced, cages were kept at 33 ± 2 °C and 50–70% RH in a dark chamber. Before the trial started on Day 0, the 1 day old emerged honeybees were kept in cages for one day (from Day −1 to Day 0) and fed with a sucrose solution (50% (*w*/v)) to give them time to become accustomed to the test conditions.

Subsequently, they were fed with the treatment solutions. Five experimental replicates were performed for each treatment under study.

Tests were conducted on the individual effects of deltamethrin (Sigma Aldrich—Schnelldorf, BO, Germany) and BPF, as well as the protective impact of BPF when administered in association with deltamethrin. Two doses of deltamethrin and one of BPF were chosen for administration. The doses were chosen for preliminary studies. Specifically, deltamethrin dosages were combined with the sucrose solution at two distinct concentrations: 2.16 mg/L and 21.6 mg/L; five control groups (dose of 0 mg/mL honeybee) were also devised. BPF dosage (1 mg/kg) was evaluated in conjunction with each sublethal dose of deltamethrin in the combined deltamethrin and BPF therapy. The effects of the treatments on honeybee survival (1–72 h) and the frequency of aberrant behavior in honeybees (1–4, 24, 48, and 72 h after treatment) were recorded.

The organization of the experimental groups is summarized below:
Deltamethrin treatment 1 (DTM 1): a low concentration of deltamethrin (2.16 mg/L) in a 50% *w*/*v* sugar solution;Deltamethrin treatment 2 (DTM 2): deltamethrin at a high dose (21.6 mg/L) in a 50% *w*/*v* sugar solution;Bergamot polyphenolic fraction treatment 1 (BPF-1): a BPF (1 mg/kg) combination in a 50% *w*/*v* sugar solution with the lower dose of deltamethrin (2.16 mg/L);Bergamot polyphenolic fraction treatment 2 (BPF-2): a combination of BPF (1 mg/kg) in a 50% *w*/*v* sugar solution with a higher concentration of deltamethrin (21.6 mg/L);Bergamot polyphenolic fraction (BPF): BPF (1 mg/kg) dose in a 50% *w*/*v* sugar solution;Control treatment: sucrose solution (50% *w*/*v*).


### 2.1. BPF Preparation

For the study, fruits of the *Citrus bergamia* Risso and Poiteau varieties were collected in the area between Bianco and Reggio Calabria (Calabria Region, Italy). Subsequently, squeezing was carried out to extract the juice from the peeled citrus fruits.

The juice underwent oil fraction depletion by stripping, clarification by ultra-filtration, and loading onto an appropriate polystyrene resin column capable of absorbing polyphenol chemicals with molecular weights between 300 and 600 Da (Mitsubishi).

A solution of 1 mM KOH was used to elute the polyphenol fractions. To lower the amount of furocoumarin, the basic eluate was incubated on a rocking platform. According to the quantity of furocoumarin impurities, the shaking duration was modified. The phytocomplex left over from the procedure used to extract furocoumarins was then neutralized by filtering cationic resin at an acidic pH. To obtain BPF powder, it was vacuum dried and then chopped to the appropriate particle size [48]. The BPF was subjected to physical, microbiological, and compositional analyses. In particular, the power was analyzed in search of flavonoids, furocoumarin, and other polyphenols. The amount of polyphenols in BPF powder was analyzed using HPLC.

### 2.2. Feeding Solutions

Deltamethrin and BPF stock solutions were created using deionized water and kept at a temperature of 4 °C. By combining the stock solutions of deltamethrin and BPF with 50% (*w*/*v*) water sugar solutions, the final treatment solutions were created.

The BPF treatment had concentrations of 1 mg/kg. Deltamethrin treatments were prepared at concentrations of 2.16 mg/L and 21.6 mg/L. The above detailed deltamethrin concentrations were mixed with BPF concentrations to verify the protective effect of the polyphenolic extract. The honeybees received feeding solutions ad libitum from the beginning of the experiment and were supplied with new ones every 24 h. At least once every three days, the feeding dilutions were made, carefully wrapped in aluminum foil to shield them from light deterioration, and kept at a temperature of 6 to 2 °C. No precipitation was ever seen in feeding solutions.

### 2.3. Behavior

The OECD’s established guidelines were used to categorize and quantify behavioral disorders [58]. Depending on the pesticide dosage, the percentage of honeybees displaying aberrant behaviors across time (1, 2, 4, 24, 48, and 72 h after treatment) and the number of improperly behaving honeybees per cage were evaluated. The following behaviors were taken into account: a curved-down abdomen, hyperactivity, apathy, motion coordination impairments, and moribundness [17]. These types of anomalous behaviors are based on ecotoxicological recommendations. We monitored each honeybee for 6 s (up to 120 s for a cage with 20 honeybees), with the cage serving as the unit of replication.

### 2.4. Food Composition

The cages were equipped with feeders. The feeders were inserted horizontally into the bottom of the cage. The feeders were sterile 2.5 mL disposable syringes with capped ends. A notch was made at the top of the syringe, near the capped end, creating a slit through which the bees fed. Food consumption was recorded daily.

### 2.5. Data Analysis

The statistical analysis was conducted using GraphPad PRISM software (version 9.0, GraphPad Software Inc., La Jolla, CA, USA). To estimate the survival function from the data obtained, the Kaplan–Meier estimator was employed. Once the analysis was obtained, the Longrank test was performed to compare the survival of the different treated groups. The difference between the experimental groups was considered significant for *p*-values of ≤0.05.

## 3. Results

### 3.1. Bergamot Polyphenolic Fraction (BPF) Analysis

The results of the analysis, including the chemical characteristics, active ingredients, heavy metals, and microbiological evaluation, are shown in Table 1 [59,60].

Toxicological tests ruled out the existence of any harmful substances. Microbiological standard testing found no bacteria or mycotoxin. Neoeriocitrin (370 ppm), naringin (520 ppm), and neohesperidin (310 ppm) were the primary flavonoids found in BPF.

### 3.2. Honeybee Survival and Mortality

Results for honeybee survival are depicted in Figure 1.

In the control groups, survival remained essentially unchanged over the three days of the trial, suffering a slight deflection on the third day. The same trend was found for the groups treated with BPF, deltamethrin 2.16 mg/L, and deltamethrin 2.16 in association with BPF. The deaths that occurred in the above mentioned groups can be associated with a physiological death that was not statistically significant when compared to the control group. Since deltamethrin 2.16 mg/mL and BPF1 were found to be non-toxic, the combination was also found to be non-toxic and safe.

Honeybees treated with deltamethrin (21.6 mg/L) had a survival probability of 38% compared to the control (sugar syrup) at 1 day of treatment. There was therefore a reduced survival, which was statistically significant (*p* ≤ 0.001) compared to the control group, already from day one of treatment. Even at day two, the difference between the two groups was statistically significant (*p* ≤ 0.001); on day three, in the deltamethrin-treated group, all subjects had died. The groups fed BPF plus deltamethrin (21.6 mg/mL) after one day of treatment had a survival probability of 86%. At two days post-treatment, the survival probability was 72%, and at three days, it stood at 46%. Therefore, the combination with BPF statistically increased the survival of honeybees compared to those in the group fed deltamethrin at 21.6 mg/mL (*p* ≤ 0.001).

### 3.3. Abnormal Behavior

The overall percentage of abnormal behavior throughout the duration of the experiment was taken as the basis for recording this data. Figure 2 depicts the trend in abnormal behavior for the various experimental groups.

The BPF, deltamethrin 2.16, and deltamethrin 2.16 in combination with BPF groups did not present significant abnormal behaviors compared to the control group. Subjects in the deltamethrin 21.6 groups exhibited the highest percentage of abnormal behavior, involving precisely 88% of all subjects under treatment. The data was statistically significant compared to the control group (*p* ≤ 0.001). BPF not only increased survival but also reduced the percentage of abnormal behaviors exhibited in a statistically significant manner compared to the deltamethrin 21.6 group (*p* ≤ 0.001).

### 3.4. Feeding Solution Consumption

The results relating to the average consumption of the experimental groups recorded in the three days of treatment are depicted in Figure 3.

The consumption was calculated by dividing the quantity of solution consumed by the number of surviving bees. Except for the BPF group, all experimental groups drank less solution than the control. The deltamethrin 21.6 group consumed food solution significantly less (*p* ≤ 0.05) compared to deltamethrin 2.16. The deltamethrin 2.16 group, in association with BPF, resulted in significantly higher food consumption than the deltamethrin 2.16 group (*p* ≤ 0.001). The groups with deltamethrin 21.6 in combination with BPF consumed more solution than both the control and deltamethrin 21.6 groups, with a statistical significance of less than 0.05 and 0.001, respectively.

## 4. Discussion

Beekeepers often utilize artificial food in the form of candies or syrup during the winter or to encourage the queen to oviposit. When the addition of food supplements to the normal diet is not dictated by management practices, they are dutifully necessary in ecosystems increasingly modified by man. Overbuilding, intensive agriculture, and increasingly monocultured land reduce the availability of food for honeybees, resulting in a shortage of nectar and pollen. Therefore, nowadays, honeybees’ normal diet is often altered. The majority of the time, the food replacements employed lack some essential nutritious elements that make up the normal diet of these pollinating insects. Along with macronutrients (carbohydrates and proteins), honeybees’ diets should also include trace elements and other substances, such as phenolic compounds, that have a significant influence on their capacity for detoxification. Phenolic compounds are among the components in honey that contribute most significantly to its antioxidant action [62]. The therapeutic qualities of honey are attributed to the phenolic chemicals found in honey, particularly the flavonoids and phenolic acid, which are important natural antioxidants of medicinal relevance [63]. Previous research has shown a considerable correlation between the antioxidant capabilities of honey derived from different floral sources and their phenolic content [64]. Numerous research investigations have also emphasized the benefits of including compounds with high nutritional value and rich in certain phenolic components. The molecular processes behind pesticide-induced toxicity entail the formation of free radicals, lipid peroxidation induction, and disruption of the body’s overall antioxidant capacity [65]. There is a strong correlation between pesticide exposure, elevated reactive oxygen species, and oxidative stress induction. According to Uchendu et al. (2018) [66], deltamethrin and chlorpyrifos (CPF), which are classified as pyrethroid and organophosphate (OP) pesticides, respectively, cause oxidative stress by changing antioxidant defense systems and producing free radicals. The authors used a combination of pyrethroid and OP insecticides, which are often applied by farmers and kept close to cereals in certain nations, such as Nigeria [66]. Rats exposed to CPF and deltamethrin, either alone or in combination, exhibited considerably higher levels of malondialdehyde and significantly lower levels of catalase, superoxide dismutase, and glutathione peroxidase than the control group [66]. Citrus fruits have been widely reported for their strong bioactivities, such as antioxidant, anti-inflammatory, antimicrobial, etc., activities [67,68,69,70]. Over 40% of BPF is made up of flavonoids; the remaining 60% is made up of various chemicals, fatty acids, carbohydrates, pectins, and maltodextrins, which are added to facilitate exsiccation [71]. The flavonoids and flavonoid glycosides present in BPF are neoeriocitrin, naringin, neohesperidin, melitidin, bruteridin, and hesperetin. These flavonoids have a wide range of positive effects on animals, including anti-inflammatory, anti-allergic, antibacterial, and anti-apoptotic capabilities. Analyzing how Naringin interacts with pesticides in animal models has drawn more attention recently. Mani et al. (2015) [72] experimented to learn more about the ameliorating benefits of naringin (100 mg/kg BW by gavage, 21 days) against hepato-pathological and hematological impairments (12.8 mg/kg BW by gavage, 21 days) in rats intoxicated by deltamethrin. The researchers discovered that the percentages of neutrophils, eosinophils, monocytes, and white blood cells in the serum, as well as the degree of tissue damage, lipid peroxidation level, lactate dehydrogenase, AST, ALT, and ALP activities in the liver of rats given deltamethrin, were significantly higher than those of the control group. Naringin therapy significantly returned all prior values to normal ranges in deltamethrin-intoxicated rats, indicating that naringin acted as a protective agent against the hematotoxicity and hepatotoxicity associated with deltamethrin exposure. Naringin’s efficacy in combating deltamethrin poisoning in rats may be attributed to its capacity to scavenge free radicals, its anti-inflammatory properties, and its immune- and defense-stimulating properties [72]. Positive results were also obtained by Agha et al. (2015) [73], who showed that Hesperidin supplementation before experimental deltamethrin intoxication drastically reduced the frequencies of chromosome aberrations and restored mitotic activity compared to the group treated with deltamethrin alone [73]. A recent study by Hybl et al. (2021) [18] showed that supplementation of phenolic acids and flavonoids to the diet of thiacloprid-intoxicated bees resulted in increased longevity, probably due to the increased detoxification capacity caused by the increased expression level of genes encoding the cytochrome P450 enzyme [18]. The BPF administered in our study could act both by attenuating the consequences of oxidative stress, such as lipid peroxidation, and by increasing the activity of the enzymatic systems deputed to detoxification. The protective results against pesticide intoxication obtained in this study with BPF were achieved in other animal species with other polyphenols of plant origin, such as quercetin, catechin, epicatechin, (−)-epigallocatechin-3-gallate, apigenin, luteolin, and taxifolin. It can be hypothesized that these other plants or their polyphenols may also have a similar protective action [74,75]. Based on the findings of this experimental study, the inclusion of phenolic compounds in the bee diet may, to a certain extent, increase the bees’ capacity for detoxification, which is frequently decreased due to malnutrition brought on by environmental degradation and the ensuing loss and contamination of food resources, as well as by factors related to the routine management of beekeeping. The survival data that were recorded in the groups in which deltamethrin was associated with BPF are promising and encourage the integration of polyphenolic substances into the normal diet. In the present study, two doses of deltamethrin were used in connection with preliminary toxicity studies. The dose chosen of 21.6 mg/L was the one that allowed us to cause the death of caged subjects within 48 h, as required by the guidelines for the evaluation of acute toxicity. The lowest dose of 2.16 mg/L was chosen because it was the one that resulted in mild toxic effects, such as behavioral abnormalities. This dose allowed us to appreciate the protective effects of BPF against mild poisoning. The intoxicated subjects with the highest concentration of the pesticide managed to survive one day longer than those who received deltamethrin alone. On the second day, all the honeybees that had been intoxicated with deltamethrin (21.6 mg/mL) had died, while in the intoxicated groups to which BPF had been added, the honeybees remained alive until the third day, with a survival rate of 46%. There was no statistically significant difference regarding survival and mortality data between the control group and the group that had been fed BPF alone. BPF, therefore, proves to be safe. The improvement trend observed with survival is also common with anomalous behaviors. Intoxicated subjects administered BPF show a lower percentage of behavioral anomalies in both groups in which BPF was supplemented at the highest and lowest doses than in the respective groups without BPF supplementation. These described results are preliminary and need to be further developed to give more strength to our evidence; however, it can be asserted that the integration of compounds with high biological value into food substitutes could help honeybees respond better to pesticide-induced stress.

## 5. Conclusions

The use of pesticides in agriculture is a common practice for the control of insects, pests, and plant pathogens. Often, the compounds used do not have good selectivity for the target but can also harm insects valuable to ecosystems, such as honeybees. In such conditions, the discovery of mixtures and/or compounds that can protect or aid the detoxification of the organism is particularly important. In this study, the BPF was shown to increase the probability of survival of honeybees intoxicated by deltamethrin. BPF may reduce intoxication damage caused by exposure to this pesticide, probably by improving the body’s detoxifying abilities. Subsequent studies are needed to better define the mechanisms by which BPF exerts its protective action. However, it can already be said that the addition of BPF as a dietary supplement could be of interest in improving the fitness of hives.

## Figures and Tables

**Figure 1 animals-13-03764-f001:**
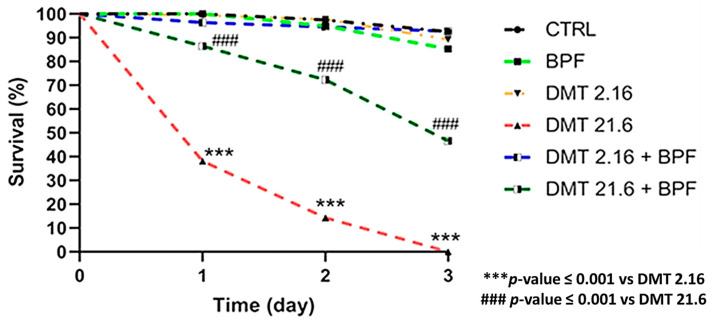
Honeybee survival curve.

**Figure 2 animals-13-03764-f002:**
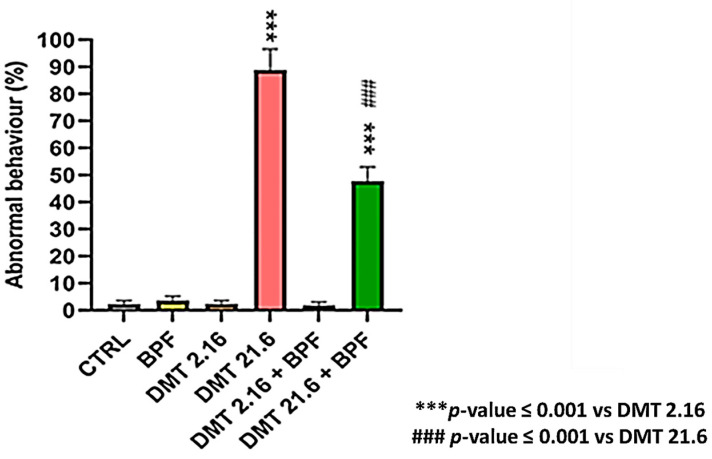
Abnormal behavior.

**Figure 3 animals-13-03764-f003:**
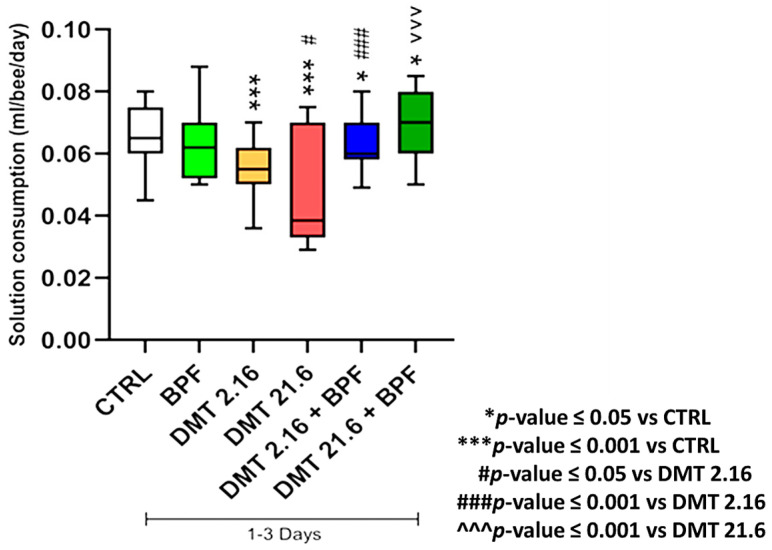
Consumption of feeding solutions.

**Table 1 animals-13-03764-t001:** Results of the bergamot polyphenolic fraction (BPF) analysis.

Description	Specifications	Methods
Chemical Characteristics		
pH	3.0–4.0	
Average Mesh Size	Pass 70 mesh	IM (0.5% in water) at 25 °C
Mass Density	30–70 g/100 mL	Sieve: (CQ-MO-023)
Water Content	<10.0%	PT CHIM 65 rev 0 2011
Organic Solvent Residue	None	ISTISAN 96/34
Soluble in 40 °C H_2_O	Good	GC: (CQ-MO-168)
Soluble in 50% H_2_O + EtOH	Good	visual: (CQ-MO-148)
Active Ingredient Strength	HPLC	visual: (CQ-MO-148)
Pesticides Residue	Negative	PT CHIM 69rev 02 011
Active Ingredients	Unit	Range
Polyphenols (neoeriocitrin, naringin, neohesperidin, melitidin, bruteridin, and hesperetin)	%	38%
Heavy Metals		
Arsenic	ppm	<2.0
Lead	ppm	<2.0
Heavy Metal Tot. Quantity	ppm	<20.0
Microbiological Evaluation		
Aerobic Plate Count	<1000 CFU/g	ISO 4833-1:2013 [61]
Yeast and Mold Count	<100 CFU/g	ISO 21527-1:2008 [61]
*E. coli*	Negative	ISO 16694-2:2001 [61]
Coliform	Negative	ISO 4832:2006 [61]
*Salmonella*	Negative	UNI EN ISO 6579:2000 [61]
*Staphilococcus aureus*	Negative	UNI EN ISO 6888-2:2004 [61]
Streptococci	Negative	UNI EN ISO 7218:2007— PT BAT26 rev0 02012 [61]
Product Treatment		
Extraction Solvents	Water + KOH	
Drying Method	Spydry	

## Data Availability

The data are kept at the University of Magna Græcia of Catanzaro and are available upon request.

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
