# Peer review of "Protective Role of Bergamot Polyphenolic Fraction (BPF) against Deltamethrin Toxicity in Honeybees (Apis mellifera)"

_animals, 2023, doi:10.3390/ani13243764_

Round 1
Reviewer 1 Report
Comments and Suggestions for Authors
Protective role of polyphenolic bergamot fraction (BPF) against deltamethrin toxicity in honey bees (Apis mellifera)
The paper contributions and its strengths
Various stressors to honey bee (Apis mellifera L.) have been investigated and found to be possible cause of the global decline of pollinators: parasites and pathogens, pesticides, climate change and nutrition. Much of the evidence collected in recent years suggests that a combination of these factors, acting in synchrony and with complex interactions, is responsible for the increased honeybee colony mortality. Pesticides are a key factor, as a multitude of studies have demonstrated their detrimental effects at both individual and colony level. Several of the most pesticides (commonly used) are systemic, protecting (and contaminating) all plant organs, including flowers, and thus nectar and pollen. Pollen is the main protein and lipid source for bee colonies and a fundamental part of the nurse bees' and larval diet, thus its contamination results in exposure of the new generation of bees, as well as the foraging and receiver bees. Nowadays, pesticides commonly used in agriculture belong to the class of pyrethroids. Deltamethrin is a Type II pyrethroid, characterized by a broad-spectrum of activity that is widely used in agriculture. Its use is not free from dangers for pollinating insects. Deltamethrin disrupts physiology in honey bees by causing memory disturbances, hypofertility, alterations in body and intestinal development, altering normal dances and foraging activity.
Furthermore, nutritional stress and malnutrition seem to be one of the primary causes of bee mortality, along with diseases. As agriculture becomes more intensive, the landscape changes and the availability of bee food supplies are progressively decreasing, which reduces environmental sustainability. Consequently, the variety of flowering plants has decreased. Many studies have demonstrated that the dietary consumption of phenolic compounds and flavonoids can increase the body's ability to respond to pesticide injuries and pathogens. Citrus Bergamia, commonly known as bergamot, has a juice and albedo, those both exhibit a distinctive flavonoid and flavonoid glycoside profile. Bergamot juice and albedo are used to produce bergamot polyphenolic extract (BPF), a polyphenol-rich fraction that has antioxidant, anti-inflammatory, lipid-lowering, and hypoglycemic properties.
In this research, the aim was to verify whether BPF could have protective effects against honey bee poisoning by the pyrethroid pesticide deltamethrin. Therefore, the highlight of this article is the use of the bergamot polyphenolic fraction.
General concept comments
Highlighting areas of weakness
The Introduction is very good and enlightening, and, the proposed aims are clear. The primary objective was to ascertain the impact of polyphenols on the mortality of honey bees poisoned by delthametrin.
However, the methodology presents methodological inaccuracies and missing controls.
Between lines 181 to 192 the five experimental groups were described:
1. Delthamethrin treatment 1 (DTM 1) a low concentration of delthamethrin (2.16 mg/L) in a 50% w/v sugar solution.
2. Delthamethrin treatment 2 (DTM 2): delthamethrin at a high dose (21.6 mg/L) in a 50% w/v sugar solution.
3. Bergamot polyphenolic fraction treatment 1 (BPF-1): a BPF (1mg/Kg) combination in a 50% w/v sugar solution with the lower dose of delthamethrin (2.16 187 mg/L).
4. Bergamot polyphenolic fraction treatment 2 (BPF-2): a combination of BPF (1mg/Kg) in a 50% w/v sugar solution with the higher concentration of delthamethrin (21.6 mg/L).
5. Control treatment: sucrose solution (50% w/v).
But, in the results, the Figure 1 (Honey bee survival curve), in line 248, shows the results of six experimental groups. In other words, the description of the Bergamot polyphenolic fraction (BPF) group was missing.
Regarding challenges and treatment: the authors must inform why they chose two acute oral doses of deltamethrin (2.16 and 21.6 mg/kg) and 1 mg/kg of BPF. Based on literature, or in acute oral toxicity, or in LD50?
An oral acute toxicity (LD50 value of 700 ng/bee) was reported (Decourtye et al. 2004). Deltamethrin can induce sublethal effects such as impaired olfaction and disturbed learning (Decourtye et al. 2004), disturbed orientation (Thompson 2003, Vandame et al. 1995), and altered foraging activity and reduced learning (Ramirez-Romero et al. 2005). Sublethal concentrations of 21.6 mg/mL (sucrose solution) deltamethrin reduced bee fecundity, decreased the rate at which bees develop into adults and increased the immature period (Dai et al. 2010).
Ramirez-Romero, R., Chaufaux, J., Pham-Delegue, M.H., 2005. Effects of Cry1Ab protoxin, deltamethrin and imidacloprid on the foraging activity and learning performances of the honeybee Apis mellifera, a comparative approach. Apidologie. 36, 601-611.
Thompson, H. M., 2003. Behavioural Effects of Pesticides in Bees–Their Potential for Use in Risk Assessment Ecotoxicol. 12, 317.
Vandame, R., Beled, M., Colin, M-E., Belzunces, L.P., 1995. Alteration of the homing-flight in the honey bee Apis mellifera L., exposed to sublethal dose of deltamethrin. Envion Toxicol Chem. 14, 855-860.
Other elucidations regarding the BPF must be made. On page 5, the table 1 shows the results of different analyzes of the bergamot polyphenolic fraction (BPF). However, these analyzes were not broken down in the methodology, such as: mass, water content, residues of organic solvents and their active ingredients, in addition to the search for toxic metals (lead and arsenic). Are these trace elements the most toxic to bees? Or, specifically, in the Calabria region, is there any industry or mining company that pollutes with these elements?
And on line 245, it was reported that no mycotoxins were found. What mycotoxins were researched? And, if they were researched, what methodology was used? Thin chromatography, HPLC, ELISA?).
The discussion should be improved and the results compared with those of other authors. Based on the mechanisms of action of deltamethrin and the type of injury that this pesticide can cause in bees, what hypothesis do the authors have for the greater survival of bees?
It is important to discuss the action of polyphenols (Neoeriocitrin, Naringin, Neohesperidin, Melitidin, Bruteridin, Hesperetin) found in the bergamot fraction.
Pay attention to the abbreviations (OP pesticides, ROS, CPF, MDA, CAT, SOD, GPx): first in full.
Reference 12 has different formatting.
12. AV, M.; Pandey, R.; Mall, P. Protecting honeybees from pesticides: a call to action. Biodiversity 2023, 24, 117–123.

Minor editing of English language required.
Author Response
Dear reviewer, thank you very much for taking the time to read our article and for your advice and comments. Below you will find the answers to your questions written in bold, point by point.
The paper contributions and its strengths
Various stressors to honey bee (Apis mellifera L.) have been investigated and found to be possible cause of the global decline of pollinators: parasites and pathogens, pesticides, climate change and nutrition. Much of the evidence collected in recent years suggests that a combination of these factors, acting in synchrony and with complex interactions, is responsible for the increased honeybee colony mortality. Pesticides are a key factor, as a multitude of studies have demonstrated their detrimental effects at both individual and colony level. Several of the most pesticides (commonly used) are systemic, protecting (and contaminating) all plant organs, including flowers, and thus nectar and pollen. Pollen is the main protein and lipid source for bee colonies and a fundamental part of the nurse bees' and larval diet, thus its contamination results in exposure of the new generation of bees, as well as the foraging and receiver bees. Nowadays, pesticides commonly used in agriculture belong to the class of pyrethroids. Deltamethrin is a Type II pyrethroid, characterized by a broad-spectrum of activity that is widely used in agriculture. Its use is not free from dangers for pollinating insects. Deltamethrin disrupts physiology in honey bees by causing memory disturbances, hypofertility, alterations in body and intestinal development, altering normal dances and foraging activity.
Furthermore, nutritional stress and malnutrition seem to be one of the primary causes of bee mortality, along with diseases. As agriculture becomes more intensive, the landscape changes and the availability of bee food supplies are progressively decreasing, which reduces environmental sustainability. Consequently, the variety of flowering plants has decreased. Many studies have demonstrated that the dietary consumption of phenolic compounds and flavonoids can increase the body's ability to respond to pesticide injuries and pathogens. Citrus bergamia, commonly known as bergamot, has a juice and albedo, those both exhibit a distinctive flavonoid and flavonoid glycoside profile. Bergamot juice and albedo are used to produce bergamot polyphenolic extract (BPF), a polyphenol-rich fraction that has antioxidant, anti-inflammatory, lipid-lowering, and hypoglycemic properties. In this research, the aim was to verify whether BPF could have protective effects against honey bee poisoning by the pyrethroid pesticide deltamethrin. Therefore, the highlight of this article is the use of the bergamot polyphenolic fraction.
General concept comments
Highlighting areas of weakness
The Introduction is very good and enlightening, and the proposed aims are clear. The primary objective was to ascertain the impact of polyphenols on the mortality of honey bees poisoned by delthametrin. However, the methodology presents methodological inaccuracies and missing controls.
1) Between lines 181 to 192 the five experimental groups were described:
- Delthamethrin treatment 1 (DTM 1) a low concentration of delthamethrin (2.16 mg/L) in a 50% w/v sugar solution.
- Delthamethrin treatment 2 (DTM 2): delthamethrin at a high dose (21.6 mg/L) in a 50% w/v sugar solution.
- Bergamot polyphenolic fraction treatment 1 (BPF-1): a BPF (1mg/Kg) combination in a 50% w/v sugar solution with the lower dose of delthamethrin (2.16 187 mg/L).
- Bergamot polyphenolic fraction treatment 2 (BPF-2): a combination of BPF (1mg/Kg) in a 50% w/v sugar solution with the higher concentration of delthamethrin (21.6 mg/L).
- Control treatment: sucrose solution (50% w/v).
But, in the results, the Figure 1 (Honey bee survival curve), in line 248, shows the results of six experimental groups. In other words, the description of the Bergamot polyphenolic fraction (BPF) group was missing.
R: Thank you very much for pointing out this oversight. We have added the sixth group in the Materials and Methods section.
2) Regarding challenges and treatment: the authors must inform why they chose two acute oral doses of deltamethrin (2.16 and 21.6 mg/kg) and 1 mg/kg of BPF. Based on literature, or in acute oral toxicity, or in LD50?
An oral acute toxicity (LD50 value of 700 ng/bee) was reported (Decourtye et al. 2004). Deltamethrin can induce sublethal effects such as impaired olfaction and disturbed learning (Decourtye et al. 2004), disturbed orientation (Thompson 2003, Vandame et al. 1995), and altered foraging activity and reduced learning (Ramirez-Romero et al. 2005). Sublethal concentrations of 21.6 mg/mL (sucrose solution) deltamethrin reduced bee fecundity, decreased the rate at which bees develop into adults and increased the immature period (Dai et al. 2010).
Ramirez-Romero, R., Chaufaux, J., Pham-Delegue, M.H., 2005. Effects of Cry1Ab protoxin, deltamethrin and imidacloprid on the foraging activity and learning performances of the honeybee Apis mellifera, a comparative approach. Apidologie. 36, 601-611.
Thompson, H. M., 2003. Behavioural Effects of Pesticides in Bees–Their Potential for Use in Risk Assessment Ecotoxicol. 12, 317.
Vandame, R., Beled, M., Colin, M-E., Belzunces, L.P., 1995. Alteration of the homing-flight in the honey bee Apis mellifera L., exposed to sublethal dose of deltamethrin. Envion Toxicol Chem. 14, 855-860.
R: Thank you for your comment. In the manuscript, explanatory sentences on the chosen doses have been added, which can be found in the 'Materials and methods' and 'Discussion' sections. The chosen dose of 21.6 mg/L was the one that allowed us to better verify what we wanted to prove experimentally. This dose caused the death of the group of caged bees within 48 hours, as recommended by the guidelines useful for determining acute oral toxicity in honey bees: 'Standard methods for toxicology research in Apis mellifera - https://doi.org/10.3896/IBRA.1.52.4.14'. On the other hand, the lowest dose (2.16 mg/L) was chosen because it was the one that could have mild toxic effects, such as causing behavioural abnormalities, but not death.
There are no published studies on the dose of the polyphenol fraction of bergamot to administered to honey bees; the dose used was chosen on the basis of preliminary studies and the experience of the group of pharmacologists involved in the research who have published numerous papers on the pharmacological activities of the polyphenol fraction of bergamot (Fitoterapia. 2011, Apr;82(3):309-16; Journal of Biological Regulators and Homeostatic Agents 2017, 31(4):1087-1093; Nutrients 2018, 10(9), 1305; Nutrients 2018, 10(11), 1604; Pharmaceuticals 2021; 14(10), 975; Nutrients 2022, 14(3), 477; Nutrients 2023, 15(6), 1334, etc.).
3) Other elucidations regarding the BPF must be made. On page 5, the table 1 shows the results of different analyzes of the bergamot polyphenolic fraction (BPF). However, these analyzes were not broken down in the methodology, such as: mass, water content, residues of organic solvents and their active ingredients, in addition to the search for toxic metals (lead and arsenic). Are these trace elements the most toxic to bees? Or, specifically, in the Calabria region, is there any industry or mining company that pollutes with these elements?
R: Thanks for your comment. We have added more information to the table. In the Bergamot (Citrus bergamia) cultivation area of ​​the Calabria region there are no industries or companies that pollute the environment with heavy metals. The analyzes were carried out according to the European Regulation 1881/2006 recently repealed by Regulation 2023/915. You can find it at the following link: https://eur-lex.europa.eu/eli/reg/2006/1881/oj. In our opinion, the total amount of heavy substances is important information that ensures that BPF does not contain toxic elements unrelated to citrus fruits, which could influence the tests. Furthermore, we already want to demonstrate the harmlessness of BPF in the event of any field studies and therefore that BPF comes into contact with the food matrices of the hive.
4) And on line 245, it was reported that no mycotoxins were found. What mycotoxins were researched? And, if they were researched, what methodology was used? Thin chromatography, HPLC, ELISA?).
R: Thank you for your comment. Mycotoxins were tested by HPLC. Following the indications of European Regulation 1881/2006 (https://eur-lex.europa.eu/eli/reg/2006/1881/oj), mycotoxins such as aflatoxins and ochratoxins were searched for. For further information on characterisation, please refer to the supplementary material in the article Pharmaceuticals. 2021; 14(10):975, as per the supplementary material available online at https://www.mdpi.com/article/10.3390/ph14100975/s1. The same information can be found in the supplementary material of the article J Tradit Complement Med. 2020 Feb 8;10(3):268-274. These two references have been added.
5) The discussion should be improved and the results compared with those of other authors. Based on the mechanisms of action of deltamethrin and the type of injury that this pesticide can cause in bees, what hypothesis do the authors have for the greater survival of bees?
R: Thank you for your comment which helps us to improve the overall quality of the manuscript. We have supplemented the discussions with hypotheses to support our results.
6) It is important to discuss the action of polyphenols (Neoeriocitrin, Naringin, Neohesperidin, Melitidin, Bruteridin, Hesperetin) found in the bergamot fraction.
R: Thank you for your comment. We have improved the discussions with the elements to clarify the function of the polyphenol fraction of bergamot.
7) Pay attention to the abbreviations (OP pesticides, ROS, CPF, MDA, CAT, SOD, GPx): first in full.
R: Thanks for this advice. We have now inserted the abbreviations in full form.
8) Reference 12 has different formatting.
- AV, M.; Pandey, R.; Mall, P. Protecting honeybees from pesticides: a call to action. Biodiversity 2023, 24, 117–123.
R: Thanks for the advice. We have corrected.

Reviewer 2 Report
Comments and Suggestions for Authors
I consider that in general the work is good, its results are useful and publishable. However, it must be improved in several aspects that are listed below.
In the introduction: I suggest updating the information on colony loss, I think it is a good justification but it must be updated and highlight the influence of pyrethroids or other pesticides on colony lossI suggest reducing the introduction, I think it is extensive and includes information that could be eliminated. for example: Bergamot is a precious product, defined by local producers as "the green gold". All information that does not have a direct relationship with the study must be deleted.
In the methodology: it must be carefully reviewed to include omitted information that does appear in the results, for example mortality, food consumption.
Line 190: any reason for these doses?
Line 238-239 This corresponds to methodology
Line 226: 120 seconds
Line 248: figure not figures
In the discussion:
what is ? ROS, CPF and MDA CAT, SOD, and GPx
It is necessary to discuss previous results on the variables: consumption, mortality, behavior, dosage, other plants
Author Response
Dear reviewer, thank you very much for the time dedicated to reading the manuscript and for the information provided. We have answered your questions point by point. The answers are written in bold.
I consider that in general the work is good, its results are useful and publishable. However, it must be improved in several aspects that are listed below.
1)In the introduction: I suggest updating the information on colony loss, I think it is a good justification but it must be updated and highlight the influence of pyrethroids or other pesticides on colony lossI suggest reducing the introduction, I think it is extensive and includes information that could be eliminated. for example: Bergamot is a precious product, defined by local producers as "the green gold". All information that does not have a direct relationship with the study must be deleted.
R: Thank you for your suggestion. We have corrected accordingly; the introduction has been streamlined by removing redundant information.
2) In the methodology: it must be carefully reviewed to include omitted information that does appear in the results, for example mortality, food consumption.
R: Thank you for this comment which helps us to improve the overall quality of the manuscript. The requested information has been added in the “Materials and Methods” section and is now highlighted in the text.
3) Line 190: any reason for these doses?
R: Many thanks for your question. There are no published studies on the doses of the polyphenol fraction of bergamot to administered to honey bees; the dose used was chosen on the basis of preliminary studies and the experience of the group of pharmacologists involved in the research who have published numerous papers on the pharmacological activities of the polyphenol fraction of bergamot (Fitoterapia. 2011, Apr;82(3):309-16; Journal of Biological Regulators and Homeostatic Agents 2017, 31(4):1087-1093; Nutrients 2018, 10(9), 1305; Nutrients 2018, 10(11), 1604; Pharmaceuticals 2021; 14(10), 975; Nutrients 2022, 14(3), 477; Nutrients 2023, 15(6), 1334, etc.).
4) Line 238-239 This corresponds to methodology
R: Thank you for your suggestion. We have corrected it by moving the sentence to the Material and Methodology section.
5) Line 226: 120 seconds
R: Thanks for the advice. We have corrected.
6) Line 248: figure not figures
R: Thanks for the advice. We have corrected.
7) In the discussion: what is? ROS, CPF and MDA CAT, SOD, and GPx
R: Thanks for the advice, we have now inserted the abbreviations in full.
8) It is necessary to discuss previous results on the variables: consumption, mortality, behavior, dosage, other plants
R: Thanks for the comment, we have integrated as indicated.
